# Differences in Reward Sensitivity between High and Low Problematic Smartphone Use Adolescents: An ERP Study

**DOI:** 10.3390/ijerph18189603

**Published:** 2021-09-13

**Authors:** Xinmei Deng, Qiufeng Gao, Lijun Hu, Lin Zhang, Yanzhen Li, Xiangyu Bu

**Affiliations:** 1School of Psychology, Normal College, Shenzhen University, Shenzhen 518060, China; xmdeng@szu.edu.cn (X.D.); ljhu94@163.com (L.H.); zhanglin09052021@126.com (L.Z.); 2Department of Sociology, Law School, Shenzhen University, Shenzhen 518060, China; liyanzhen2018@email.szu.edu.cn (Y.L.); buxiangyu2017@email.szu.edu.cn (X.B.)

**Keywords:** problematic smartphone use, reward sensitivity, adolescents, event-related brain potentials (ERPs)

## Abstract

Background: Problematic smartphone use is highly prevalent in adolescent populations compared to other age groups (e.g., adults and young children). Previous studies suggested that higher levels of reward sensitivity were associated with problematic smartphone use. Therefore, the current study investigated the neural processing of monetary and social reward and punishment feedbacks between high and low problematic smartphone use adolescents. Methods: 46 adolescents participated in the current study and they were categorized into two groups based on their level of problematic smartphone use: those who obtained low scores on the measure of problematic smartphone use were categorized as Low Problematic Smartphone Use (LPSU), and those who obtained high scores on the measure of problematic smartphone use were categorized as High Problematic Smartphone Use (HPSU). Electrocortical activities were recorded during the processing of monetary and social reward and punishment feedback. Results: (1) LPSUs evoked larger P3 in the social punishment condition than in the monetary punishment condition. HPSUs evoked larger P3 in the social reward condition than in the monetary condition. (2) The feedback-related negativity (FRN) amplitudes in the reward condition were significantly larger than those in the punishment condition. (3) HPSUs induced larger reward positivity in social feedback conditions than in monetary feedback conditions, while there were no significant differences between the two types of conditions in the LPSUs. Discussion: The results provide neural underpinning evidence that high sensitivity to social rewards may be related to problematic smartphone use in adolescence.

## 1. Introduction

Adolescence characterizes increasing demands of creating social connections, mainly due to heightened autonomy and independence [1]. Under the background of the popularity of online social networking, adolescents show high rates of dependence on mobile phones, especially smartphones, and there is a high incidence of problematic smartphone use among adolescents [2]. Evidence from several countries showed that the proportion of problematic smartphone users among adolescents was more than 20% [3]. Previous studies often conceptualized problematic smartphone use as the excessive and compulsive use of smartphones, with social or occupational impairment, including dependence and symptoms seen in addictive disorders, such as withdrawal and tolerance [4]. In addition, problematic smartphone use leads to psychiatric and psychological difficulties in adolescents, such as nomophobia, depression, anxiety, and low self-esteem [5,6,7]. Thus, what causes this problematic smartphone use and how to diminish the overuse of smartphones in teenagers have been widely addressed by parents, educators, and researchers.

One mechanism that is typical in the development of addictive behaviors is sensitivity to rewards. Contingency between action and follow-up reward and non-reward is critical to shaping behavior [8]. Children develop an understanding that some stimuli (or situations) are related to rewarding experiences, while others are related to non-rewarding or even punishing experiences [9]. Smartphones are full of attractive reward cues (e.g., social reward cues, such as likes on Facebook or WeChat; non-social reward cues, such as gold coins in internet games), leading to addictive behaviors. Therefore, increased sensitivity to the rewards in smartphone activities might render some individuals more vulnerable than their peers to problematic smartphone use.

Problematic smartphone use is generally considered to be a behavioral addiction, and may be related to similar neuropsychological (i.e., craving and tolerance) and personality characteristics (i.e., reward-seeking and impulsivity) to other addictions, especially the most similar behavioral addiction, problematic internet use (PIU). A growing number of studies on PIU have focused on the relationship between the rewarding mechanism and the development and maintenance of its addiction behavior. These studies showed that deficit in the rewarding mechanism plays a crucial role in the development and maintenance of PIU because it induces “reward bias” for potential reward cues, such as video game stimulation or online gambling cues [10]. Furthermore, cross-sectional studies using mixed methods, such as self-report [11], event-related potentials [10,12], and fMRI [13], revealed that internet addicts have enhanced reward sensitivity and decreased loss sensitivity compared to participants in the control group.

Previous studies, however, show inadequate understanding of problematic smartphone use because of several limitations. First, existing research focused mainly on PIU. Therefore, little is known about the relationship between reward sensitivity and mobile phone addiction, which is similar but different from PIU. For example, internet game addiction is the most common subtype of PIU. However, problematic smartphone use has mainly focused on social network overuse [14]. Second, previous research [12,13] has also centered primarily on the neural basis of non-social reward processing in neuroscience, generally using monetary reward tasks (i.e., doors task and guessing task), and relatively little attention has been paid to reward sensitivity related to decision making in social settings [15]. In addition, prior studies have mainly focused on college students, and little is known about the reward mechanism of technology addiction behaviors in other periods of adolescence. The present study examined how problematic smartphone use is related to electrocortical activities during the processing of different types (social vs. non-social rewards) of reward and punishment feedbacks in different periods during adolescence.

### 1.1. Problematic Smartphone Use and Reward Sensitivity in Adolescents

Adolescence is a period with heightened sensitivity to social rewards [16] compared to non-social rewards. Older adolescents pay more attention to social evaluations than monetary rewards than their younger counterparts because social rewards have greater reinforcement values in this age [9]. Adolescents show hypersensitivity to both positive and negative social stimuli and for social rewards and social punishments alike [17]. Others’ emotional expressions are more salient and distracting for adolescents than for adults or children because these symbols of social approval are particularly rewarding to adolescents. Moreover, adolescents tend to spend more time in social interaction with their peers than with their families.

There is great significance in examining reward sensitivity during adolescence. Neuroimaging studies suggested that when receiving social approval, adolescents showed more activations than adults in the ventral striatum, medial prefrontal cortex, and orbitofrontal cortex, which are associated with rewarding stimuli [18]. Another longitudinal study found increased activation of adolescents in the ventral striatum when passively viewing positive and negative faces from age 10 to age 13 [19]. In addition, adolescents showed more activation in the anterior cingulate cortex (ACC) than adults. Furthermore, when receiving rewards, adolescents (ages 12–17 years) showed increased striatal activation relative to pre-adolescents (ages 7–12 years) and adults [20]. Taken together, because of the susceptibility to social influence and the increased activation in brain regions to social stimuli, the reward sensitivity of adolescents may have significant impacts on their daily behaviors.

Besides reward sensitivity, adolescence is also a life period heightened in addictive behaviors [21]. Prior studies have suggested that, compared to people in other age groups, adolescents are vulnerable to addictive behavior because of how they experience rewards [5,22]. For example, problematic smartphone users showed a higher impulsivity level and sought immediate rewards [23]. Increasing research has suggested that problematic smartphone use is closely affected by the motivational mechanisms related to reward sensitivity and approach tendencies [5]. Neuroimaging research also suggested that the brain’s reward circuitry also plays a role in smartphone activities such as social media. For instance, receiving “Likes” on social media photographs is related to increased activation in the dorsal and ventral striatum, which is implicated in reward processing [24]. Furthermore, the level of activation in the ventral striatum is related to the level of social media use [25]. The imbalance between the highly activated reward-related brain region and protracted matured control regions may lead to more impulsive and problematic smartphone use. Taken together, these suggested that the relatively high reward sensitivity of adolescents could be one of the possible reasons leading to the susceptibility of problematic smartphone use.

### 1.2. Biomarkers of Reward and Punishment Feedbacks

Electrophysiological evidence shows that feedback processing is a dynamic process that can be indexed by different feedback-locked event-related potentials (ERPs). For example, feedback-related negativity (FRN) was elicited at the frontocentral recording sites around 250 ms following feedback. It is sensitive to the valence of feedback (reward vs. punishment) and the amount of reward (small vs. large) [26]. FRN can also predict subjective emotional experiences after receiving contextual social evaluation [27]. Importantly, FRN is associated with evaluating the meaning of the emotional motivation of the feedbacks [28]. The emotional motivation hypothesis is one of the theories to study the internal nervous mechanism of FRN [29]. It points out that FRN reflects the process of evaluating the meaning of emotional motivation in a feedback stimulus and emphasizes the importance of emotional motivation [30]. Therefore, the higher the motivation and input, the higher the attention to feedback.

As an essential component for feedback, P3 is a positive component of 300–400 ms after presenting the feedback results [26]. Compared to low reward cues, high reward cues activated higher P3 amplitude [31]. These increased P3 values contributed to the increased motivational salience of rewards and the updating of the internal environment. Sometimes, this reflects valence when defined in high-level affective evaluations, but not about the reward value per se [32]. Wu and Zhou [33] pointed out that P3 was related to attention distribution and emotional intensity during reward feedback.

Another one of the most studied components related to reward processing is reward positivity [34]. Reward positivity was elicited about 240–340 ms post feedback over frontal–central areas of the scalp [35]. Thus, it reflects an initial evaluation of information that predicts forthcoming rewards and the subsequent reevaluation of that information during reward processing.

Differences between ERPs elicited by positive and negative feedback resulted from neural modulation of the reinforcement learning system [36]. For example, previous research found increased FRN following unexpected social feedback and P3 following positive social feedback [37]. FRN may represent a motivation towards negative context and punishment avoidance, whereas P300 may reflect a post-feedback evaluation, with a proactive control to support future behavior and increase rewards, related to approach motivation [38]. Reward positivity is more sensitive to win feedback than no-win feedback. Additionally, approach-motivated states promote performance monitoring, as reflected by the reward positivity for rewarding outcomes [39].

The ERP markers of interest (P3, FRN, and reward positivity) are also associated with the cognitive and neural processes of addictive behaviors. For example, NoGo-P3 amplitude indicated lower inhibitory control in participants with PIU compared to their peers without PIU [40]. The relatively worse response inhibition ability of the addictive group might derive from the early stage of selected attention to the later stage of successful cognitive control. Additionally, individuals with high scores on the Internet Additive Test (IAT) showed reduced FRN and increased P300 amplitude for stimuli represented as more salient and rewarding [12]. P3 responses to reward conditions were significantly correlated with problematic internet use [41]. Additionally, approach-motivated goal pursuit promotes performance monitoring, as reflected by the reward positivity for rewarding outcomes [39]. Approach-motivated goal pursuit could encourage attaining desired objects or goals, which is the core of addictive behaviors. Previous research suggested that the increased amplitudes of reward positivity resulting from repeating positive performance feedback could be important evidence of the reinforcement learning framework [42]. Furthermore, the dysfunction of the reinforcement learning framework could be the possible reason for addictive behaviors. Therefore, the high sensitivity of ERPs (P3, FRN, and reward positivity) to reward processing may be considered a biomarker of addictive behaviors (e.g., problematic smartphone usage).

The current study examined how problematic smartphone use is related to electrocortical activity during the processing of different types of reward and punishment feedback between high and low problematic smartphone use adolescents. We predicted that problematic smartphone use adolescents would induce larger reward processing-related ERPs in social feedback conditions than in monetary feedback conditions, especially for social reward conditions.

## 2. Method

### 2.1. Participants

All the adolescents came from urban communities in a metropolitan city in China. There were 25 adolescents with low problematic smartphone use (LPSU; 11 male and 14 female, aged from 10 to 15 years old, *M*_age_ = 11.58, *SD* = 1.22) and 21 adolescents with high problematic smartphone use (HPSU; 14 male and seven female, aged from 11 to 15 years, *M*_age_ = 12.86, *SD* = 1.16). Previous studies suggested that individual reward sensitivity and addictive behaviors tend to increase during early adolescence and peak around mid-adolescence (age 14 or 15). Based on the developmental trend of reward sensitivity and addictive behaviors, we recruited participants aged 10 to 15 years old. That is, the participants in the current study covered early adolescence and mid-adolescence periods [20,21]. Participants were recruited based on their scores on the 10-item Mobile Phone Problematic Use Scale (MPPUS-10) [43]_._ The 10-item Mobile Phone Problematic Use Scale (MPPUS-10) [43] assessed adolescents’ problematic smartphone use on a 5-point Likert scale (1 = *not true at all*, 5 = *extremely true*, e.g., *“I find it difficult to switch off my mobile phone.”*). The MPPUS-10 demonstrates adequate validity and reliability [44]. Like previous research, higher MPPUS-10 scores are associated with more time spent online and higher levels of problematic smartphone use [43]. After reverse scoring the relevant items, we averaged across the 10 items to create the total score of the adolescents’ problematic smartphone use level (α = 0.869). In the present study, the participants’ scores of the MPPUS ranged from 10 to 83. The higher the MPPUS-10 score adolescents had, the higher the level of problematic smartphone use (see Appendix A). HPSUs (*M_HPSU_s* = 28.52, *SD* = 9.85) and LPSUs (*M_LPSU_s* = 57.67, *SD* = 10.76) differed significantly on the problematic smartphone use score (*p* < 0.001). A sample of this size is in line with typical ERP studies [40].

Participants were recruited via flyers that invited healthy volunteers from junior middle schools to participate in a study of smartphone use. No participant had a neurological or psychiatric disorder history, as determined by self- or parental-report. In the sample, 43.48% of the adolescents were only children and the rest had siblings. Approximately 69.57% of fathers and 65.22% of mothers had received a college education. Participants and their parents gave informed consent before participating in the study.

### 2.2. The Monetary and Social Reward Tasks

The monetary and social reward task [45] assessed the neural processing of monetary and social reward and punishment feedback between high and low problematic smartphone use adolescents. In prior research, the task was widely used to successfully assess reward sensitivity in adolescent samples [46]. There were four experimental conditions: “monetary reward” (MR), “monetary punishment” (MP), “social reward” (SR), “social punishment” (SP). The experimental conditions were presented in a pseudo-random order to prevent the same feedback conditions from appearing in the same sequence.

Each experimental trial began with a fixation cross at the center of the screen for 500 ms (see Figure 1). Next, a red gift box appeared at the center of the screen. The participants were instructed to press the space bar to react to the gift box as quickly as possible with their dominant hand to obtain positive feedback. If participants reacted before the gift box disappeared, they received money or social rewards; if they reacted after it disappeared, they failed to obtain money or gained social punishment. To prevent automated responses and ensure that the participants’ attention was focused on the upcoming target, the inter-stimulus interval (ISI) between the cue offset and the target was jittered between 200 ms and 3000 ms. Button presses during target presentation were counted as hits and resulted in positive feedback (i.e., MR and SR). Conversely, late button presses after the target had disappeared led to negative feedback (i.e., MP and SP). Then, a question mark was shown for 1000 ms. Last, the feedback was presented for 1000 ms. After receiving each piece of feedback, participants rated their feeling of pleasure from 1 to 5 (e.g., how pleasant do you feel about the result of the game?).

In the MR trial, participants were informed that each instance of positive feedback would result in monetary gain (+1 CNY note, see samples of the picture stimulations in Figure 2). In the MP trial, participants were informed that each instance of negative feedback would result in a failure to obtain money. In the SR trial, participants were shown a picture of a smiling human face. In the SP trial, participants were shown a picture of a frowning human face. As to the facial photographs, the images on the money note in the monetary trials were different from the images of smiling faces in the SR trial. All facial photographs were from the native Chinese Facial Affective Picture System [47] and checked through a predictive test of liking, familiarity, and intimacy.

The monetary and social reward experiments task consisted of 8 blocks. Each block had 40 experimental trials that consisted of four types of feedbacks in a pseudo-random order. E-Prime software was used to present all stimuli against a black background on a 21-inch monitor, with a viewing distance of approximately 80 cm. An experimental session took 30 min for each participant.

### 2.3. Procedure

We employed a 2 (feedback valences: reward vs. punishment) × 2 (feedback types: monetary vs. social) × 2 (group: high vs. low problematic smartphone use adolescents) mixed measures design, with problematic smartphone use as the between-subjects factor and feedback valence and feedback types as within-subjects factors.

Participants completed the demographic questionnaire and Mobile Phone Problematic Use Scale (MPPUS-10) before the formal experiment. Then, the participants completed the monetary and social reward task in the sound attenuation and electromagnetic shielding EEG lab. Trained graduate research assistants described the procedures to participants before each experimental session and thoroughly debriefed the participants after the experiment. To encourage participants’ involvement in the reward task, the research assistant told participants that they would receive actual monetary rewards according to their performance in the task.

### 2.4. Psychophysiological Recording, Data Reduction, and Analysis

Continuous EEG signals were recorded with a 64-channel amplifier (BrainAmp, Brain Products, Physio-tech Co., Ltd., Tokyo, Japan) based on the 10/20 system, with two electrodes placed on the left and right mastoids. The EEG was sampled at 500 Hz, and impedance was kept below 5 kΩ. The data were referenced offline to the averaged mastoid references, and the bandpass was filtered from 1 Hz to 30 Hz [48]. Eye movements and blink artifacts were corrected using the independent component analysis (ICA) algorithm implemented in Brain Vision Analyzer 2.0 (Brain Products, Physio-tech Co., Ltd., Tokyo, Japan). The data were segmented in epochs from 200 ms before stimuli onset until 1500 ms after the onset. The ERPs were constructed by separately averaging them according to the four experimental conditions. For each ERP average, the baseline level set is 200 ms before feedback. Trials with artifacts exceeding ±100 μV were excluded from further analysis. The mean number of valid epochs averaged per condition was from 37.00 (92.50%) to 37.97 (94.93%) for the HPSUs. The mean number of valid epochs averaged per condition was 33.16 (82.90%) to 34.59 (86.48%) for the LPSUs. Repeated measures ANOVAs were conducted to rule out the possible impact of the number of valid epochs on the neural findings. Results indicated no significant effects of the valid number of neural findings on the neural findings (all *p* > 0.05).

Based on the existing literature and visual inspection, in the present study, P3 was evaluated at Pz as the highest amplitude between 250 ms and 350 ms following stimulus onset [49]. FRN was evaluated at FCz as the highest amplitude between 200 ms and 300 ms following stimulus onset [50]. Reward positivity was evaluated at FCz as the average difference waves (reward–punishment) between 250 ms and 350 ms following stimulus onset [39]. (Behavioral data were examined using a 2 (feedback type: social feedback vs. monetary feedback) × 2 (group: LPSUs vs. HPSUs) repeated measures ANOVA. The results showed that the main effect of feedback type was not significant, *F*(1,44) = 1.854, *p* = 0.180, η_p_^2^ = 0.040. The main effect of group was also not significant, *F*(1,44) = 0.027, *p* = 0.870, η_p_^2^ = 0.001. The interaction of feedback type × group was not significant, *F*(1,44) = 2.628, *p* = 0.112, η_p_^2^ = 0.056. In addition, correlation analysis between behavioral data and EEG data was performed and found that there was no significant result between behavioral data and EEG data (*p*s > 0.05). In this study, there was no significant result of the behavioral data and a significant relationship between behavioral data and EEG data. Therefore, behavioral results were not included in the main text.) Behavioral ratings of emotional experience, P3, FRN and reward positivity in the different experimental conditions between HPSUs and LPSUs were examined using repeated measures ANOVAs. Fisher LSD test was used for multiple post hoc comparisons. Greenhouse–Geisser corrections were applied to p values in multiple-df comparisons.

Before conducting repeated measures ANOVAs, Shapiro–Wilk tests were conducted to examine if the dependent variable conforms to a normal distribution. Results of the Shapiro–Wilk tests showed that most of the examined dependent variables followed normal distribution (*p* > 0.05), except the scores of the reward positivity in monetary feedback condition among LPSUs (*p* < 0.05). Therefore, logarithmic transformations of the scores were performed before the repeated measures ANOVAs and we found that this confirmed a normal distribution after the logarithmic transformations (*p* > 0.05).

## 3. Results

### Neural Results

Average amplitudes of different components between different conditions are shown in Table 1 and Figure 3.

***P3 (Figure 4)*.** The main effect of group was not significant, *F* (1,44) = 0.633, *p = 0*.431, *η_p_^2^* = 0.014. The main effect of feedback types was significant, *F* (1,44) = 19.904, *p* < 0.001, *η_p_^2^* = 0.311. The P3 amplitudes of monetary feedback were lower than those of social feedback. The main effect of feedback valence was not significant, *F* (1,44) = 0.8^40^, *p = 0*.365, *η_p_^2^* = 0.019. The two-way interaction of group and feedback types was not significant, *F* (1,44) = 0.113, *p* = 0.738, *η_p_^2^* = 0.003. The two-way interaction of group and feedback valence was not significant, *F* (1,44) = 0.398, *p* = 0.532, *η_p_^2^* = 0.009. The two-way interaction of feedback types and feedback valence was not significant, *F* (1,44) = 0.430, *p* = 0.515, *η_p_^2^* = 0.010. However, the three-way interaction of group*feedback types*feedback valence was significant, *F* (1,44) = 6.168, *p* = 0.017, *η_p_^2^* = 0.123. Post hoc tests found that LPSUs evoked larger P3 in the SP condition than in the MP condition, *F* (1,44) = 12.891, *p* = 0.001, *η_p_^2^* = 0.227. HPSUs evoked larger P3 in the SR condition than in the MR condition, *F* (1,44) = 12.995, *p* = 0.001, *η_p_^2^* = 0.228.

***FRN (Figure 5).*** The main effect of group was not significant, *F* (1,44) = 0.618, *p* = 0.436, η_p_^2^ = 0.014. The main effect of feedback types was not significant, *F*(1,44) = 1.675, *p* = 0.202, η_p_^2^ = 0.037. The main effect of feedback valence was significant, *F*(1,44) = 5.124, *p* = 0.029, η_p_^2^ = 0.104. The FRN amplitudes in the reward condition were significantly larger than those in the punishment condition. None of the two-way and three-way interactions were significant (all *F* ≤ 2.042, *p* ≥ 0.160, η_p_^2^ ≤ 0.044).

***Reward Positivity (Figure 6)*.** The main effect of group was not significant, *F* (1,44) = 1.289, *p* = 0.262, η_p_^2^ = 0.028. The main effect of feedback types was not significant, *F* (1,44) = 1.726, *p* = 0.196, η_p_^2^ = 0.038. The interaction of group and feedback types was significant, *F* (1,44) = 5.660, *p* = 0.022, η_p_^2^ = 0.114. Post hoc tests found that HPSUs induced larger reward positivity in social feedback conditions than in monetary feedback conditions (*F* (1,44) = 6.272, *p* = 0.016, η_p_^2^ = 0.125), while there were no significant differences between the two types of condition in the LPSUs (*F* (1,44) = 0.622, *p* = 0.435, η_p_^2^ = 0.014).

## 4. Discussion

The present study investigated how problematic smartphone use is related to electrocortical activities during the processing of monetary and social rewards and punishment feedbacks between adolescents with high and low problematic smartphone use. We found that HPSUs evoked larger P3 in the social reward condition than in the monetary condition compared to LPSUs, suggesting larger sensitivity differences to reward versus punishment in the HPSUs compared to the LPSUs. Additionally, HPSUs induced larger reward positivity in social feedback conditions than in monetary feedback conditions compared to LPSUs. The findings were consistent with the previous notions that problematic smartphone use might be related to highly activated reward-related brain regions [25]. Furthermore, because of the higher reinforcement values of social rewards, increased activation in the dorsal and ventral striatum might reflect adolescents’ emphasis on social evaluations and increase the level of social media use [25]. However, it is worth noting that the relationships between HPSUs and social issues may be related to one recently widely concerned phenomenon termed Fear of Missing Out (FoMO). It is defined as a particular kind of anxiety that arises from a person’s fear of missing out compared to others who might be having rewarding experiences from which one is absent; FoMO is characterized by the desire to stay continually updated on what others are doing [51]. Moreover, we also found that the withdrawal factors of the MPPUS-10 are anxiety reactions similar to FoMO. Therefore, the unsatisfied social relatedness needs that are entailed in FoMO could motivate one to engage in problematic internet use, especially in problematic social networking use. These results paralleled the findings from addiction studies.

Differences between ERPs elicited by different types of feedback in the present study might elucidate problematic smartphone users’ cognitive and neural processes. We found that social feedback but not FRN evoked larger P3 wave and reward sensitivity in HPSUs than it did in LPSUs. Previous studies suggested that FRN is related to punishment avoidance in the behavioral inhibition system (BIS), whereas P3 is related to approach motivation in the behavioral approach system (BAS) [38]. Reward positivity to rewarding outcomes could also indicate approach motivation [39]. Both P3 and reward positivity indicate active approach-motivated goal states that stem from the reward-related brain system that drives individuals to attain a desired outcome [39]. Consistently, in our study, adolescents with higher P3 and reward positivity when receiving social feedback also reported more severe cases of problematic smartphone use. Thus, higher P3 and reward positivity may represent higher approach motivations towards social feedback in adolescents with problematic smartphone use.

Previous research showed that puberty had its most significant impact on the emotional aspects of adolescent life [52]. Smartphones offer pleasurable emotional experiences that potentially function as rewards, which might increase the chance that process-oriented smartphone use develops into habitual use [53]. The unique neural mechanism underlying emotional reward processing in adolescents makes this age group especially vulnerable to problematic smartphone use.

Besides approach-motivated goal states, social purposes also influence habitual smartphone use. For example, receiving “likes” on social media can increase the activation of the reward network in the human brain [54]. Social rewards gained on social media enable adolescents to construct self-identity and form positive self-evaluation. However, excessive smartphone usage may compromise adolescents’ interpersonal relationships in the real world [55]. Therefore, adolescents’ neural and social development makes the pursuit of social rewards via smartphone usage not only pleasant but also meaningful.

We also found that adolescents’ FRN amplitudes in reward conditions were significantly larger than those in punishment conditions. Previous studies have shown that FRN primarily reflects the difference between good and bad outcomes and unexpected and expected feedbacks [56]. However, according to the emotional motivation hypothesis, FRN represents emotional motivation towards feedback stimuli [30]. Therefore, the higher the motivation towards stimuli, the more attention is given to feedback. In addition, neural activities in the ventral striatum, activated upon receiving rewards, are enhanced during adolescence [20]. This enhancement in neural activities could be a possible reason for the higher FRN amplitudes among our participants in the reward conditions.

There were practical implications of the main results in this study. The excessive pursuit of social rewards in the virtual world may cause reduced social interactions in the real world [57]. Research shows that counseling services that emphasize face-to-face social interactions can help adolescents with addictive smartphone use problems because real-world face-to-face interactions can also generate pleasant emotional experiences and a sense of self-approval. The pleasant emotional experiences and self-approval from social rewards during smartphone use can also be derived from face-to-face interaction. Thus, adolescents who are in the situation of addictive smartphone use can receive the help of individual or group counseling to improve their face-to-face interaction [57]. Besides, an increase in the time allocated to real-world social relations provides opportunities for adolescents to fulfill their social needs and increase perceived social support [58]. Our ERP results indicated that the attainment of social rewards via smartphone use created pleasant feelings and a sense of self-value. Therefore, intervention programs for adolescents with addictive smartphone use should consider their need for self-construction.

The current study revealed different ERP activities among participants in different reward conditions. This finding suggests that problematic smartphone use might be associated with different underlying neural activities during different stages of reward processing.

One of the present study’s major limitations was our relatively small sample size. Although our sample size is in line with typical ERP studies [59], it is necessary to include more participants to gain larger statistical power in future studies. Additionally, it would be feasible to include more adolescents with different levels of problematic smartphone use in the study to increase the generalizability of our findings. Another limitation of our study was that the questionnaire measured the extent of problematic smartphone use via the self-report method. Even though our result was consistent with existing research showing that higher MPPUS-10 scores are correlated with more time spent online and higher levels of problematic smartphone use [43], the self-report measure that we used may be affected by subjective factors. Therefore, it is necessary to use a more reliable and objective tool to measure problematic mobile phone use in future studies. Finally, regarding individual differences, although we controlled participants’ demographic variables during data analysis, we did not measure any personality variables. Therefore, another important future direction is identifying how personality differences could moderate the relationship between problematic smartphone use and reward sensitivity. In the present study, we used facial images to provide feedback in both the monetary and social trials in the experiment. However, the images on the money note in the monetary trials were different from the images of smiling faces in the SR trial. Although monetary feedback is tangible, images of smiling faces are not tangible in the same way. In the future studies, it is necessary to examine how facial images in the monetary and social trials influence the findings of the study.

## 5. Conclusions

In summary, this is the first ERP study to examine the relationship between problematic smartphone use and reward sensitivity in adolescents. Since adolescents experience increasing demand for creating interpersonal interactions and social connections, they are more sensitive to social rewards [16]. Our study consistently observed a higher P3 and reward sensitivity during the processing of social feedback in adolescents who had higher scores in problematic smartphone use. Due to the social development of the brain, expectations and reinforcements from social interactions increase from pre-adolescence to early adolescence. Although behavioral measures suggest that social rewards (e.g., receiving a “Like” from others) might facilitate habitual smartphone use, few ERP studies have examined this notion with a sample of adolescents. These findings from the study deepened our understanding by exploring the neural and behavioral effects of social rewards in smartphone use during this particular age period. The high sensitivity to social rewards identified in the problematic smartphone usage group in this study could provide ideas for the prevention and remission of smartphone addiction in adolescents.

## Figures and Tables

**Figure 1 ijerph-18-09603-f001:**
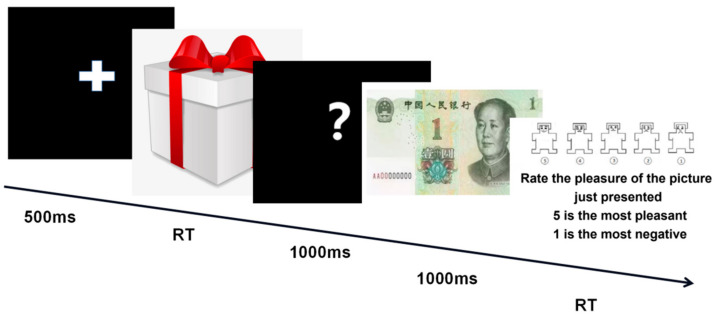
The procedure of each trial.

**Figure 2 ijerph-18-09603-f002:**
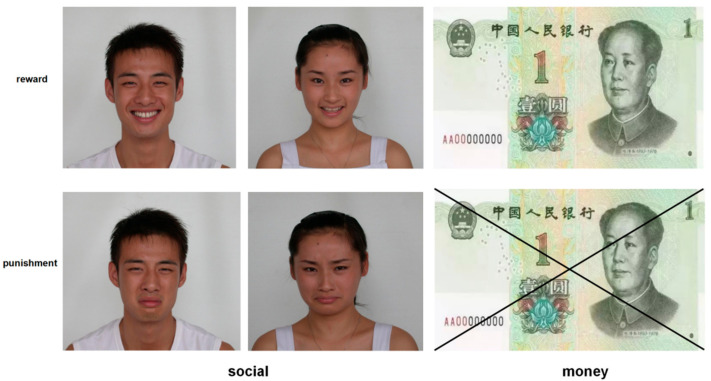
Picture stimulations.

**Figure 3 ijerph-18-09603-f003:**
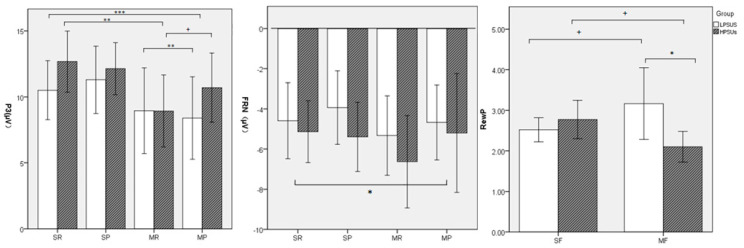
Average amplitudes of different components between different conditions (P3, FRN, and reward positivity). Note. SR = social reward; SP = social punishment; MR = money reward; MP = money punishment. Note: * *p* < 0.05; ** *p* < 0.01; *** *p* < 0.001.

**Figure 4 ijerph-18-09603-f004:**
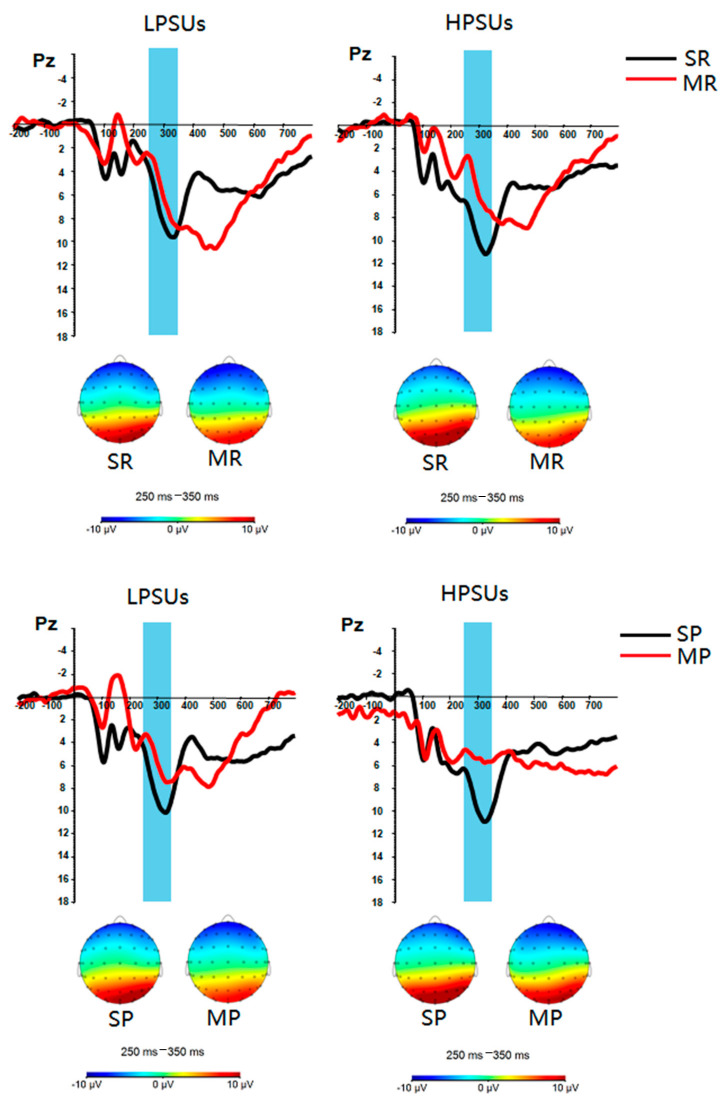
Stimulus-locked grand averaged waveforms and scalp distributions of the highest amplitude in response to different conditions at Pz. SR = social reward, MR = monetary reward, SP = social punishment, MP = monetary punishment.

**Figure 5 ijerph-18-09603-f005:**
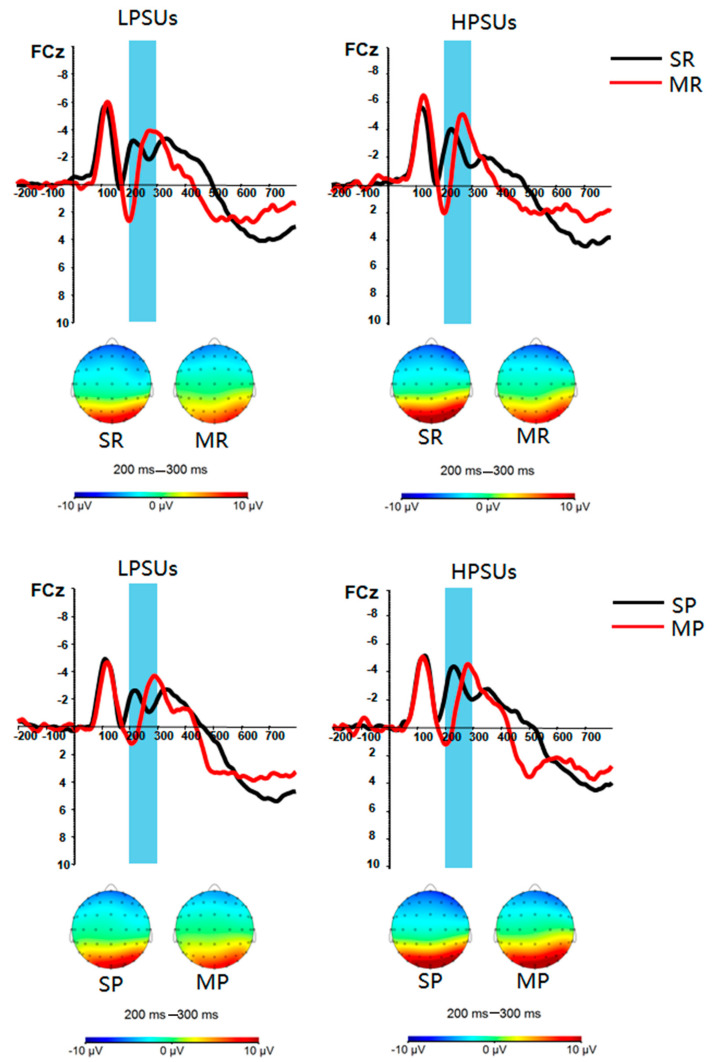
Stimulus-locked grand averaged waveforms and scalp distributions of the highest amplitude in response to different conditions at FCz. SR = social reward, MR = monetary reward, SP = social punishment, MP = monetary punishment.

**Figure 6 ijerph-18-09603-f006:**
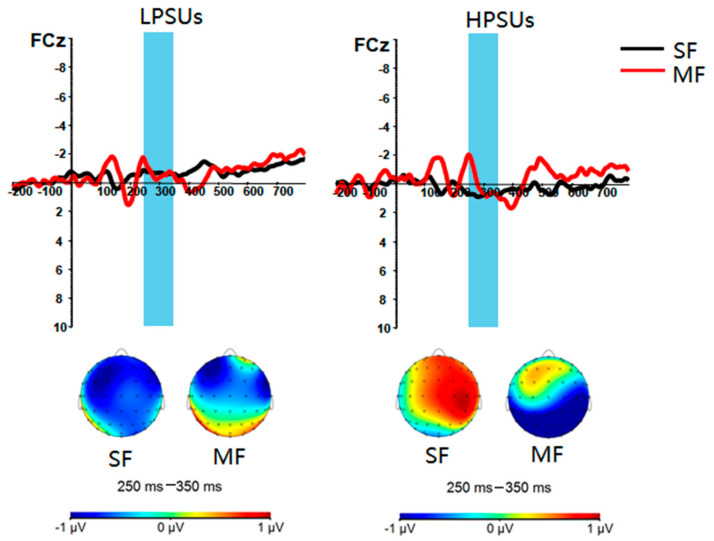
Stimulus-locked grand average difference waves (reward–punishment) and scalp distributions of the highest amplitude in response to different conditions at FCz. SF = social feedback, MF = monetary feedback.

**Table 1 ijerph-18-09603-t001:** Average amplitudes of P3, FRN, and reward positivity between high and low problematic smartphone use adolescents in different conditions.

ERPs	Conditions	LPSUs	HPSUs	*t*	*d*	*p*	95% CI	
P3	SR	10.51 ± 5.41	12.68 ± 5.08	−1.39	−0.42	0.17	−5.31	0.97
	MR	8.95 ± 7.87	8.93 ± 6.00	0.01	0.00	0.99	−4.20	4.25
	SP	11.29 ± 6.18	12.14 ± 4.33	−0.53	−0.16	0.60	−4.07	2.39
	MP	8.39 ± 7.56	10.70 ± 5.74	−1.15	−0.35	0.26	−6.36	1.75
FRN	SR	−4.59 ± 4.58	−5.14 ± 3.40	0.45	0.14	0.66	−1.89	2.98
	MR	−5.32 ± 4.79	−6.63 ± 5.05	0.90	0.27	0.37	−1.62	4.24
	SP	−3.94 ± 4.44	−5.40 ± 3.79	1.19	0.36	0.24	−1.02	3.94
	MP	−4.67 ± 4.53	−5.20 ± 6.50	0.32	0.10	0.75	−2.76	3.81
RewP	SF	0.88 ± 0.29	0.95 ± 0.39	−0.64	−0.20	0.52	−0.27	0.14
	MF	0.97 ± 0.61	0.66 ± 0.43	1.94	0.59	0.06	−0.01	0.63

Note: SR = social reward, MR = monetary reward, SP = social punishment, MP = monetary punishment, SF = social feedback, MF = monetary feedback.

## Data Availability

The data used in this study are available upon reasonable request from the corresponding author.

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
