# Peer review of "Differences in Reward Sensitivity between High and Low Problematic Smartphone Use Adolescents: An ERP Study"

_ijerph, 2021, doi:10.3390/ijerph18189603_

Round 1
Reviewer 1 Report
Thank you for the opportunity to read an interesting study. The publication is part of the current research on the issue of problematic smartphone use using EEG techniques. The study was conducted on a small sample (46 people) divided into two groups: low PIU and high PIU. The conclusion of the study is interesting and significantly linked to the social etiology of PIU. However, given the importance of the data collected, I have some questions and suggestions:
The introduction lacks a definitional anchoring of problematic smartphone use. It would be useful to add a clear definition and show to what extent problematic internet use is connected to (is part of) other phenomena such as: nomophobia, FOMO.
The term Internet addiction disorder appears in the text. However, it should be stressed that this concept is denied in the literature. Some authors (e.g. Zsolt Demetrovics) emphasise that there are no clear diagnostic criteria for IAD so far, so other terms that do not generate stereotypes should be used.
In the methodological note, information should be added in which country the research was carried out. Information on the year of the study is also missing. This is important in the context of the covid pandemic, which has significantly changed the style of new media use among adolescents.
Is the use of ANOVA justified given the distribution of responses (normal distribution)?
The MPPUS-10 tool used is based on self-declarations. This is a major limitation in the context of measuring risk behaviour mediated by new media. On the one hand, the authors use "hard" experimental procedures. On the other hand there is a "soft" tool based on self-evaluation among adolescents. This is a disadvantage when it comes to triangulating highly reliable data with subjective data.
The issue of generalisation resulting from sampling should be clearly highlighted in the article. This is a major limitation. However, it is known that experimental studies are based on smaller samples than typical diagnostic studies (showing only the scale of the phenomenon). However, it is worth mentioning this fact in the text.
The discussion explaining the relationship between high levels of problematic smartphone use and social issues is not sufficiently carried out. There is a lack of in-depth interpretation of this fact. Could it also be due to other related phenomena such as FOMO? Which factor from the tool used connects most with social issues? It is worth deepening this thread.
The research is interesting but requires some additions. I am keeping my fingers crossed for corrections to be made.
Reviewer 2 Report
This paper investigates the neural reward system (using EEG) in adolescents with and without problematic smartphone use.
Although the topic is interesting, I have significant concerns about missing information. There are also significant issues of English grammar throughout that should be edited for clarity.
Major concerns are noted below:
- It is unclear why the authors used such a large age-range for the study (10-15 years of age). This age range includes children who have not undergone puberty and those who likely have already experienced puberty. This is a potential confound and should be considered when analyzing data. For example, age could be used as a covariate in analyses.
- The authors do not note what scores on the MPPUS mean (e.g. what scores are considered problematic vs not)
- There are insufficient details about the experimental design. How long did participants have to react to the box in order to get positive feedback?
- Relatedly, why did the authors use a reaction-time paradigm rather than more traditional reward paradigms?
- The authors cite a previous study for the use of the task in the current study. However, the cited study used an entirely different design/paradigm. It is therefore unclear how/why/from where the authors got this paradigm.
- The figure of the task design is insufficient rendering quality to accurately see what is being displayed on screen. Also, there is no image showing the positive/negative social feedback.
- It is a confound to have the nonsocial condition include tangible rewards (e.g. money) and the social condition to have non tangible rewards (e.g. smiling faces). This should be considered when interpreting the results.
- The authors note in the results that "the highest amplitude in the 200ms picture onsets served as the baseline". It is not clear what that means. Typically, the 200ms prior to feedback onset serves as a baseline.
- It appears that significantly less trials were included for the LPSU group. Were the number of included trials compared between groups?
- It appears as though the authors did not include data about behavioral ratings from participants about the rewards. Why was that data not included?
- For the P3, it appears that the nonsocial condition elicited a significantly later P3. That is, the amplitude differences appear largely driven by the chosen ms time window, especially for the LPSU group in the reward condition.
- For the reward positivity, the authors consider marginally significant results to be significant, and do not appear to report all of the follow-up tests for the significant interaction effect
- The discussion requires extensive English grammar editing for clarity/readability.
Round 2
Reviewer 1 Report
The article has been improved. The authors have responded in great detail to all the issues raised in the first review. The text is structured correctly. The results are interesting. The only weakness of the text is the small research sample.
In its present form, the study can be recommended for publication.
Reviewer 2 Report
the authors have responded to my concerns. However, within the edited text there are some remaining language issues (e.g it says that in the monetary punishment condition participants were told they could win money, which is not correct).
Also, it is important for the authors to acknowledge that while money is tangible (i.e. money can be taken after the experiment), images of smiling faces are not tangible in the same way. It would be helpful for this to be considered as a limitation.
